# Hydrothermal Synthesis and Characterization of Zeolite A from Corn (Zea Mays) Stover Ash

**DOI:** 10.3390/ma14174915

**Published:** 2021-08-29

**Authors:** Norway Pangan, Susan Gallardo, Pag-asa Gaspillo, Winarto Kurniawan, Hirofumi Hinode, Michael Promentilla

**Affiliations:** 1Arts and Sciences Department, Technological University of the Philippines, Taguig 1630, Philippines; 2Chemical Engineering Department, De La Salle University of the Philippines, Manila 0922, Philippines; susan.gallardo@dlsu.edu.ph (S.G.); pag-asa.gaspillo@dlsu.edu.ph (P.-a.G.); michael.promentilla@dlsu.edu.ph (M.P.); 3Department of International Development Engineering, Tokyo Institute of Technology, Tokyo 152-8550, Japan; kurniawan.w.ab@m.titech.ac.jp (W.K.); hinode.h.aa@m.titech.ac.jp (H.H.)

**Keywords:** hydrothermal synthesis, corn stover ash, zeolite A, calcination, alkalinity, curing time, cation exchange capacity

## Abstract

This study deals with the impact of calcination, alkalinity, and curing time parameters on the hydrothermal synthesis of zeolite A. The zeolite A sample, produced from corncob-stalk-and-leaves (corn stover) ash was characterized using X-ray diffraction (XRD), Fourier transform infrared spectroscopy (FT-IR), thermo-gravimetric analysis (TGA), and scanning electron microscopy (SEM). The results showed that calcination, alkalinity, and curing time have significant effects on the crystallization and the morphology of zeolite A. In addition, these parameters also impacted the cation exchange capacity. Furthermore, the synthesized zeolite A was obtained using a calcination temperature of 500 °C within two hours of airflow, which is much lower than the temperatures previously reported in the literature for an agricultural waste and other waste materials. A fusion ratio of corn stover ash:NaOH of 1.0:1.5 and a curing time of nine hours were achieved. This is a major result as this curing time is much lower than those featured in other studies, which can reach up to twenty-four hours. In this paper, cubic crystal with rounded edge of zeolite A, having a cation exchange capacity of 2.439 meq Na^+^/g of synthesized anhydrous zeolite A, was obtained, which can be a good candidate for ion-exchange separation.

## 1. Introduction

The Philippines is recognized for its agricultural contributions. In spite of the aim to create a highly urbanized and industrialized country out of the archipelago, a major part of the Philippine economy still relies on agriculture as the people’s primary livelihood. People in the rural areas of the country still support themselves through agriculture—farming, livestock, forestry, and fisheries. Statistics show that these four sub-sectors of agriculture together represent 38.9 percent of labor employment, thereby contributing twenty percent of the country’s gross domestic product (GDP) [1,2]. Corn is the second leading Philippine agricultural yield next to rice, with one third of Philippine farmers depending on corn as their significant source of income. Reports state that corn is one major staple substitute in cases of rice scarcity, and is a primary source of animal industry feed in the country. Corn is also gaining popularity in terms of usage in manufacturing [3].

It is thus safe to say that corn is an important agricultural product in the country. Corn could therefore possibly create mounting agricultural residues; it is estimated that the Philippines can produce a million ton of corn cobs annually [4,5]. The chief components of a corn are the corn grain and corn stover (the corn cob, stalk, and leaves). The 1:1 ratio of grain:stover accounts for a slight rise in the weight of the stover in the total biomass produced by a given corn farm. The burning process usually carried out in a corn farm, producing a corn stover as a by-product in which the corncob, stalk and leaves are burned at temperatures of 800 °C to 1000 °C. Corn stover ash is of a spongy and porous structure by nature. Twenty percent of the weight of a corn stover is represented by the corn cob [6] and consists of a huge amount of lignocelluloses materials. Some farmers use corn stover as feeds for their animals; others burn these as wastes. The ash produced after burning requires definitive and permanent disposal. The main chemical composition of corncob-stalk-and-leaves (corn stover) ash is silica. The synthesis of zeolites, which are obtained from the combustion of ash particles, would therefore be an ideal application in this area, since these ash materials are inexpensive and available in abundance.

Generally, published works [7,8,9] have revealed that the major silica precursors used are commercially-made alkoxysilane compounds such as sodium silicate, tetraethyl orthosilicate (TEOS), and tetramethyl orthosilicate. However, Nakashima et al. [10] stated that close contact with a certain type of alkoxysilane compounds could lead to death. Numerous researchers have reported that it is expensive to produce synthetic zeolites from alkoxysilane chemical sources; however, commercial silica is available in different forms (amorphous solid, fume, gel, and sol) and is found to be cheaper but has unpredictable reactivity and selectivity [11]. Thus, there is a need to look for an ecological, inexpensive, and non-toxic silica precursor. Naturally occurring silicas found in agricultural waste are a good substitute for commercial silica precursors. Thus, raw materials from agricultural waste, such as corncob-stalk-and-leaves ash [12], rice husk ash [13,14,15,16,17,18,19,20,21,22,23,24], sugar cane bagasse ash [25,26], barley ash [27], coal ashes [28,29,30,31,32,33,34], clay minerals [35,36,37,38,39,40], industrial slags [41], natural zeolites [42], and solid waste incineration ashes [43,44,45], are used as alternative and cheap source materials for the synthesis of zeolite. The use of agricultural waste materials, such as those in the field of water purification and in the removal of ammonium or heavy metals, makes them appealing as useful products that contribute to the mitigation of environmental problems. They are even more appealing now that the limited capacity of available landfills have caused environmental and social problems, arising from the growing amount of solid waste from the day-to-day living of the population.

Several studies have been carried out using the above-mentioned agricultural waste materials for the synthesis of zeolite. However, no studies have been conducted using corn stover ash in synthesizing zeolites, particularly zeolite A. This study specifically deals with synthesizing zeolite A from corn stover ash—specifically synthesizing zeolite A that shows the Linde type A (LTA) structure. The zeolite A pore has three-dimensional aluminosilicates that are perpendicular to each other in the x, y, and z planes. Its structure contains pores running perpendicular to each other in the x, y, and z planes, with an eight-member oxygen ring diameter that is small, at 4.2 Å, and a minimum of 11.4 Å larger free cavity diameter [46]. The square-faced cubic structure surrounding the cavity is connected by eight sodalite cages. The zeolite A is fascinating due to its useful spacio-specific catalysis super-cage structure. There is a large enough structure in the inner cavity for varying reactions to take place, while only a specific structure can get into the small pore [47,48].

Zeolites are increasingly popular in terms of their numerous technical applications, including as adsorbents, catalysts and ion exchangers, and separators [18,26,44,49]. The traditional applications of zeolite A for alternate zeolite production are as follows—the zeolite, when synthesized, can be applied as a precursor for making other zeolite materials such as zeolite N [50] and SSZ-13 [51]; for anti-bacterial and medical purposes such as the atomic resolution imaging of silver exchange zeolite A [52] and as a mineral-based hemostatic agent [53]; for catalytic processes such as the selective catalytic reduction of NO_x_ [54,55], the production of biodiesel [56], oxidation reactions [57], and nitrous oxide (N_2_O) decomposition [58]; and for membrane separation. In addition to these applications, zeolite A is employed in the development of inorganic membranes to replace organic membranes, which has garnered popularity because of its potent benefits in terms of structural stability even when heated, more sturdy mechanical strength, and much improved resistance to chemicals, such as water permeability [59] and natural gas dehydration [60]. It is also used for water softening applications—used as softener for reverse osmosis brine or simulated seawater [61] and is also utilized in the detergent industry to improve the removal of both calcium and magnesium [62]. The most significant characteristic of zeolite A is the fact that it is now widely used in ion exchange separation, for example, in wastewater treatment, where zeolite A is found to have a variety of applications in ion exchange or the adsorption of range water contaminants such as ammonium ions [63], lead, copper [64], and nickel ions [65], as well as malachite green dyes [66] and oil-water [67].

The chemical and physical properties of zeolite A are largely related to their morphology, structure, and size distribution, and they also have significant effect on their applications and properties. Recent developments in this area have shown that zeolite A’s mechanism of nucleation and the parameters of hydrothermal synthesis have a definite impact on the size of the crystal and the morphology of zeolite A, for example, the calcination, alkalinity and curing time that are utilized during the hydrothermal synthesis. Research results have shown that zeolite A morphology results from the reaction mixture of the chemical composition, specifically alkalinity. In addition to this, the preceding studies indicated that the above-stated parameters have caused the transformation phase of zeolite A to sodalite.

The current work pertains to the synthesis of zeolite A from corn stover ash that was revealed to be a source of silica when the amorphous form was extracted through the milling, calcining, and acid leaching of ash. A close study is essential in order to determine the effects of these parameters’ (such as calcination, alkalinity, and curing time) on the cation exchange capacity, morphology, and particle size for hydrothermal synthesis. Likewise, characterizations of corn stover ash were investigated, including elemental analysis, X-ray fluorescence (XRF), X-ray diffraction (XRD), Fourier transform infrared spectroscopy (FT-IR), thermo-gravimetric analysis (TG-DTA), and scanning electron microscopy (SEM).

## 2. Materials and Methods 

### 2.1. Sources of Raw Materials

The corn stover ash from a corn producer in Ilocos Norte, Luzon, Philippines, was used as a raw material. The mechanical, thermal, and acid treatments are considered the pre-treatment processes of corn stover as shown in Figure 1. Corn stover was milled to particle sizes smaller than 0.0741 mm during the mechanical treatment. Next, the corn stover was calcined at an air flow rate of 100 mL/min at various temperatures (300, 400, 500, 600, 700 °C) for two hours to eliminate any incorporated organic matter for the thermal treatment. Finally, an acid-leached treatment was applied, with a 1.0 M HCl solution at 100 °C for an hour to the calcined corn stover. The solid/liquid ratio of calcined ash to acid was 1.0 g/5 mL. The metallic impurities in agricultural waste can be reduced to negligible concentrations by treating with hydrochloric acid [68,69].

All chemicals were directly used as received without any further purification and are of analytical grade, obtained from Wako Pure Chemical Industries, Ltd., Japan. The aluminum source for zeolite A synthesis was sodium aluminate (0.78 molar ratio of Al/NaOH). The sodium source used was sodium hydroxide powder (97.0%). Hydrochloric acid (1 M) was used for acid treatment.

### 2.2. Methods

A two-step zeolite A preparation method was employed as illustrated in Figure 2, which encapsulates the scheme that consists of silica extraction and zeolite A production.

#### 2.2.1. Silica Extraction from Corn Stover Ash

The first preparation step was the extraction of the silicon and aluminum content from corn stover particles. A fine fraction of pre-treated (milled, calcined, and acidified) corn stover was mixed with sodium hydroxide powder of various weight ratios of corn stover ash: NaOH, 1.0:0.5, 1.0:1.0, 1.0:1.5, and 1.0:2.0 and heated at 300 °C for two hours to achieve a fused powder. This was cooled at room temperature and milled. The fused powder and deionized water mixture have a weight ratio of fused mass:deionized water of 1:2 and aged for two hours at room temperature with stirring. After which, the suspension was subjected to a centrifuge process then sieved to eradicate solid residue to achieve clear supernatant. Measurement of concentrations of Si, Al, and Na in the supernatant were made using inductively coupled plasma-atomic emission spectroscopy (ICP-AES SPS 7800) (SII Nano Technology Inc., Tokyo, Japan).

#### 2.2.2. Zeolite A Preparation

The zeolite A synthesis was prepared using the silicon and aluminum in the supernatant; prior to this, its concentration was first measured using ICP-AES. A high silica supernatant and deionized water mixture was made using a weight ratio of silica supernatant:deionized water of 2:1. In the preparation of the reaction mixture of Si/Al with a molar ratio of 1, the diluted silica supernatant was blended with intermittent drops of a solution of sodium aluminate made from a gram of sodium aluminate powder into 5 mL of deionized water, while under vigorous stirring. This mixing was continued for an hour, then the hydrothermal treatment was carried out using a Teflon-lined stainless-steel vessel at 90 °C for various times (6.0, 9.0, 12.0 h.) at a pressure that was autogenous. Finally, the zeolite A synthesis was filtered, washed several times with deionized water, and then dried under 100 °C for twenty-four hours.

### 2.3. Characterizations and Cation Exchange Capacity (CEC)

The elemental compositions of corn stover ash, specifically Si, Al, Mg, P, K, Ca, etc., were examined using X-ray fluorescence (XRF, Horiba XGT-7200, Horiba, Ltd., Kyoto, Japan). The N_2_ adsorptions of the corn stover ash and synthesized zeolite A were carried out using an Autosorb1C AX1C-MP-LP (Quantachrome Instruments, Boynton Beach, FL, USA) and the specific area was calculated using the BET method. The crystal phases of the powders were classified using the powder diffraction technique with a Multiplex (Rigaku, Tokyo, Japan) with Ni-filtered Cu-Kα radiation (λ = 0.15418 nm), operating at 40 kV and 20 mA over the 2-theta range of 5° to 90°. A Fourier transform infrared (FT-IR) spectrometer (Perkin Elmer, Waltham, MA, USA) equipped with a diffuse reflectance attachment was used for measuring the diffuse reflectance infrared Fourier transform spectroscopy (DRIFTS). Transmittance mode analyses were performed using the KBr pressed technique in 4000–400 cm^−1^ regions. The thermal stability of corn stover ash was carried out in air using thermogravimetry or TG-DTA Thermo plus TG 81200 (Rigaku, Tokyo, Japan) in the temperature range of 20 °C–900 °C at a heating rate of 10 °C/min. The morphologies of the corn stover ash, calcined and acid treated corn stover ash, and the obtained and commercial zeolite A were evaluated using the scanning electron microscopy (SEM VE-8800) (Keyence, Osaka, Japan) using Keyence VE-8800 with applied potentials of 5 kV and 20 kV. The samples were coated with a thin layer of gold for the SEM measurements.

Commercial and synthesized zeolite A was carried to its hydrogen form, which concurs with the standard procedure [48,70]. Under continuous stirring and at room temperature, commercial and synthesized zeolite A (one gram) was contacted for 2 h at 400 rpm with 100 mL of 1.0 M NH_4_Cl. The solid zeolite A was then separated from the liquid through filtration and again contacted with the new 1.0 M NH_4_Cl solution. This process was repeated 5 times. Consequently, the zeolite was dried at 550 °C for 3 h, and again the same process which was formerly described was repeated and the zeolite was dried. Finally, the zeolite was washed with distilled water and dried at 80 °C for 12 h. Moreover, the cation exchange capacity (CEC) of commercial and synthesized zeolites were tested using the sodium acetate method so as to exchange all the cations in the material with sodium through the use of sodium acetate and then extract all of them using ammonium acetate. The extracted sodium ions were measured using inductively coupled plasma-atomic emission spectroscopy (ICP-AES) using SPS 7800 (SII).

## 3. Results and Discussion

### 3.1. Characterization of Corn Stover Ash

The chemical composition of the corn stover ash was determined by XRF and was as follows: Si was 62.17 wt.%, Al was 4.72 wt.%, Mg was 5.84 wt.%, P was 6.85 wt.%, K was 10.95 wt.%, Ca was 7.47 wt.%, Ti was 0.15 wt.%, Mn was 0.05 wt.%, Fe was 1.58 wt.%, Cu was 0.01 wt.%, Zn was 0.07 wt.%, Sr was 0.02 wt.%, and La was 0.12 wt.%. The surface area of corn stover ash was 11.47 m^2^/g. As shown in Figure 3, the XRD pattern of the corn stover ash confirmed the presence of characteristic peaks of calcite (CaCO_3_), cristobalite high (SiO_2_), and quartz (SiO_2_). The broad peaks between 2θ of 20°–50° implied the presence of the amorphous phase of carbon and silica. The patterns were classified based on the characteristics of X-ray diffraction peaks with the d-values obtained with profile diffraction file (PDF) data. Raw corn stover ash (as received) matched the characteristic peaks of calcite with PDF# 47-1743 and the characteristic peaks of cristobalite high (SiO_2_) with PDF# 85-0621 and quartz (SiO_2_) with PDF# 83-0539.

Generally, the vibration of a particular configuration of chemical bonding is associated with its absorption bands. Published data from various studies [18,26,35,36,71,72] were used as a guide to interpret corn stover ash’s spectra. As expected, the corn stover’s FT-IR spectra revealed a comparable series of absorption bands; the main composition of the sample was oxide of Si. The IR absorption bands and its associations to specific vibrations were gleaned as illustrated in Figure 4 and Table 1. The peak at 1187 and 1063 cm^−1^ corresponded to the stretching vibrations of bridge bonds that were asymmetric –v_as_ T–O(T) (Si–O stretching); the 1028 cm^−1^ belonged to the bending vibrations –O–H; the peak at 881 cm^−1^ has been assigned to Si translation; the peaks at 783, 753, 710, and 650 cm^−1^ were assigned to symmetrical stretching vibrations of bridge bonds –v_s_ T–O–T (vs Si–O–Si); the peak of 581 matched the coordinated Al octahedral; and lastly, the peaks at 448, 502, and 540 cm^−1^ corresponded to bending vibrations –T–O–T or the deformation vibration of Si–O.

Figure 5 shows the SEM micrographs which illustrate the incidences of the corn stover ash achieved after pre-treatments, disclosing a noticeable modification in the morphology of the initial surface of the corn stover ash. The raw corn stover ash (as received) can be recognized by its honeycomb-like morphology (Figure 5a). Based on the SEM images in Figure 5b,c, the pre-treatments of calcining and acid leaching of corn stover ash cannot destroy the aggregation of its crystals; however, they were able to eradicate any impurities in the material, which resulted in a minor enhancement of the dispersibility of the corn stover ash crystals. The calcined-acid-treated corn stover ash honeycomb-like crystals disappeared instead, and a pile of irregular of microscopic particles was formed.

### 3.2. The Effect of Calcination Temperature

The initial step in zeolite A synthesis is the thermal treatment of the raw and purified corn stover ash to achieve a more reactive phase. Calcination is the process of removing volatile substances through the heating of solids to a high temperature; this is also known as the process of purification. It is a pretreatment given to corn stover ash with the goal of getting rid of the calcite. The calcite in the FTIR measurements shown in Figure 4 gives four peaks at 710, 753, and 783 cm^−1^, which match the symmetric in-plane bending, whereas the 881 cm^−1^ peak matches the asymmetric out-of-plane bending of CO_3_ groups. The X-ray diffraction patterns of corn stover ash after calcination at different temperatures are shown in Figure 6. The characteristic peak of calcite at a 2θ value of 29.419° was found in the diffractogram of the washed and milled corn stover ash. In the activation temperatures of 300 °C to 500 °C, there was no major structural change observed or the calcite peak did not decompose. However, when activated at 600 °C there was a slight disappearance of the calcite peak. This disappearance was made even clearer when activated at 700 °C. This result conforms with the study of Zunino et al. [73], stating that calcite decomposes between 600 °C and 800 °C.

The thermal stability of the initial corn stover was obtained via thermogravimetric analysis (TG-DTA), as shown in Figure 7. The transition of corn stover was detected near 500 °C. The loss below 350 °C corresponds to the corn stover water and the loss in the range of 350 °C–550 °C can be attributed to the dehydroxylation of the structural OH of the corn stover.

Figure 8 shows the FT-IR spectra of corn stover ash under different calcination temperatures. Comparing the IR signal spectra of the different determined thermal conditions, the presence of the Si–O bond in multiple signals in the raw corn stover ash indicates the presence of a silanol, due to the presence of alcohol. However, in all higher thermal conditions, the presence of alcohol could not be observed; hence, this could indicate siloxane functionality of the Si–O bond. In the absence of alcohol in higher thermal conditions, it could be implied that it was transformed into a carboxylic acid in higher thermal conditions in different signals; 1646 cm^−1^ at 300 °C, 1628 cm^−1^ at 400 °C, and 1646 cm^−1^ at 500 °C–700 °C, but was not present in the raw corn stover ash. This could be brought about by hydrolysis caused by calcination in higher thermal conditions, and this may also result in the presence of fewer silanols to bond to the terminal carboxylate.

All spectral signals in higher thermal conditions—500 °C, 600 °C, and 700 °C consecutively—had the same spectral reading, which may imply that by 500 °C all the decomposable functional groups under calcination had reached their satiation points and could no longer be transformed further. Hence, the presence of Si–O bonds could not be decomposed by calcination.

Generally, the calcination temperature is chosen in the range that enables the complete transformation into the amorphous phase. From the thermodynamic point of view, the higher the calcination temperature, the better transformation can occur. However, there should be some balance struck between this and the economic reason–energy and effort resources. Therefore, the calcination temperature of 500 °C was chosen for the following experiment since it is high enough for the transformation and it only uses a lower amount of electrical power. The results further showed that synthesized zeolite A was obtained using a calcination temperature of 5000 °C within two hours of airflow, which is much lower than the temperatures ranging from 550 °C–8000 °C reported in other studies of agricultural waste [17,21] and other materials [29,31,35,36,37,38,41].

### 3.3. The Effect of Alkalinity

One of the major determinant process, which is its concentration base, controls the crystallization of zeolites [29]. The alkalinity surge resulted in an increase in the rate of crystallization through crystal growth and nucleation, which are consequences of the rise in aluminate, silicate, and aluminosilicate concentrations [39,47,74,75]. The particle size decreases with this increase in alkalinity. This yields a narrow dispersal of particle size because of the rise in the rate of nucleation and polymerization between aluminate and polysilicate anions [37,76,77].

The effects of the phase composition of synthesized zeolite A material were investigated with fusion ratios of corn stover ash:NaOH from 0.5, 1.0, 1.5, and 2.0, respectively, at 500 °C with different curing times of 6, 9, 12, and 24 h. The growth of synthesized zeolite A in the samples was observed through the comparison of the d-values of the products obtained from the PDF card no. 39-0222 and the d-values of commercial zeolite A samples. The most notable change detected in the XRD diffractograms is the presence of the characteristic peaks of zeolite A. The synthesized products matched the characteristic peaks of zeolite A at 2θ values of 7.200°, 10.180°, 12.461°, 16.080°, 21.640°, 23.940°, 26.060°, 27.061°, 29.880°, 30.778°, 32.481°, 33.319°, and 34.084°, respectively.

Figure 9 illustrates the XRD patterns of the treated corn stover ash samples with 0.5 to 2.0 ratios of weight of NaOH powder; zeolite A was determined to be the chief integral mineral phase and no hydroxysodalite was observed for the 0.5 to 1.0 fusion ratios. In the 1.5 fusion ratios of the 6-, 9-, 12-h curing times, no hydroxysodalite was observed; however, in the 24-h curing time, a few traces of hydroxysodalite with PDF card no. 1-72-2329 were observed at the 2θ value of 28.22°. These few traces of hydroxysodalite were also observed in the 2.0 fusion ratio in all its curing times (9-, 12-, and 24-h), set in the two 2θ regions, where one characteristic peak corresponded to zeolite A at 2θ values of 23.96°, which intensified, thereby manifesting the combination of zeolite A and hydroxysodalite. It was also observed that in the 24-h curing time another 2θ region was manifested, with the value of 34.14°.

These results confirmed that moderate alkalinity conditions are significant in the crystallization of zeolite A. The literature revealed that any increase in the alkalinity boosts the silica source solubility. Likewise, an increase in the supersaturation degree of the hydrogel synthesis can give way to an increase in the rate of nucleation [78]. This leads to the fact that the rate of crystallization can be enhanced through an increase in alkalinity. On the contrary, alkalinity conditions that are too strong can suddenly transform synthesized zeolite A into hydroxysodalite. The fusion ratio of corn stover:NaOH of 2.0 yielded a mix of crystalline zeolite A and a few traces of hydroxysodalite. These XRD data illustrate that the intensities of the hydroxysodalite peaks in the pattern increased with the rise in the NaOH concentration. Similar observations were achieved by Andrades et al. [79], Belviso et al. [80], Cundy and Cox [81], Gougazeh and Buhl [36], Johnson et al. [40], Liu et al. [82], Mostafa et al. [83], and Youcef et al. [84]—synthesized zeolite A was achieved at concentrations that were of low base and the intensities decreased with concentrations that were of higher base.

### 3.4. The Effect of Curing Time

The effects of various curing times on the crystallization of zeolite A from corn stover ash during hydrothermal treatment were investigated. Figure 10, Figure 11 and Figure 12 show the SEM micrographs of the incidences of the zeolitic products achieved after hydrothermal synthesis of corn stover ash with fusion ratios of 0.5 to 2.0 under different curing times. Cubic rounded-edge crystals with sizes ranging from 0.6 μm to 2.0 μm after hydrothermal treatment for different curing times can be seen in the synthesized zeolite A, verifying the materialization of zeolite A that is similar to commercial zeolite A.

The SEM micrographs of the zeolites initially showed poorly crystalline zeolites. The cubic rounded-edge crystals did not manifest entirely in the 6-h curing time. However, in the 9- and 12-h curing times, we observed a similar morphology to that of synthesized zeolite A.

These results are in agreement with those of Ayele et al. [37], Ayele et al. [39], Purnomo et al. [26], and Zhang et al. [77], stating that the related actions of alkalinity and aluminum substances in the reaction mixtures are the bases of crystalline zeolite A’s morphological characteristics. The alkalinity content extensively determines the size of the zeolite A crystals. This means that, as shown in the experiments conducted, the sample size of the zeolite A crystals exhibits a marginal decrease when there is an increase in alkali fusion.

The SEM micrographs illustrated in Figure 10d, Figure 11d and Figure 12d showed no formation of spheroidal sodalite structures, as noted with increasing NaOH concentrations (fusion ratio of 2.0), which was expected. However, aggregates began to appear, which could imply the initial formation of hydroxysodalite structures. In the experimental results of this study, the observed morphologies were the same as those detailed in the previous studies [36,41,43] and this correlates with the data obtained from the XRD results, as seen in Figure 9. Thus, the crystal size was decreased; this was brought forth by the rise in alkalinity and further expounded by the increase in nucleus numbers that occurred in the gel medium. This was also shown in the results gathered by Krznarić et al. [75], Mousavi et al. [85], and Xing Dong et al. [76]. On the other hand, unlike the significant effect of alkalinity on the morphology of the synthesized zeolite A from corn stover ash, the curing time does not have a notable effect on the synthesized zeolite A particle size. This result conforms with the observations made by Abdullahi et al. [86], Bayati et al. [87], and Johnson et al. [40].

The percentages of crystallinity of the synthesized zeolite A in various curing times and fusion ratios, calculated based on XRD profiles, are summarized in Table 2. The study showed that, for the synthesis of zeolite A using corn stover ash, the optimum crystallinity of 58.18% was attained with a fusion ratio of 1.0:1.5 and a curing time of 9.0 h, which is of higher value compared with the percentage crystallinity of commercial zeolite A, which is 53.51%. Furthermore, among the percentage crystallinity values, the fusion ratio of 1.0:0.5 and curing time of 6.0 h gave the lowest value of 49.52%. Comparing the trends of the percentages of crystallinity of the synthesized zeolite A samples revealed that there are upward patterns from 0.5 to 1.5 fusion ratios in different curing times. However, increasing the fusion ratio to 2.0 revealed a downward trend in all curing times, clearly showing that an increase in alkalinity corresponds to a decrease in crystallinity. This is congruent with the theory that a smaller particle size or crystallinity percentage can be found under more alkaline conditions. Obviously, the system has an additional OH concentration, and with this, more silicate and aluminate ions can be dissolved. Thus, there is a higher possibility for the crystallization of zeolite A.

### 3.5. Cation Exchange Capacity (CEC)

The key prerequisite for the application of zeolite A as an ion exchanger is its cation exchange capacity (CEC) [39,44,88]. Studies have generally revealed that a higher alumina content inside the crystalline structure explains the higher CEC potential of zeolite A. This adds to a huge surface negative charge, which can incorporate more cations [26] and a higher sodium content [89]. It is thus safe to conclude that the forthcoming application of zeolite samples can be solved by removing Na^+^ ions and other cations from seawater. Table 3 illustrates the results obtained from this study pertaining to the removal capacity of sodium, expressed in meq Na^+^ per gram of anhydrous commercial and synthesized zeolite A in hydrogen form.

The results show that as the fusion ratio escalates, the CEC values also increase, except for the fusion ratio of 1.0:2.0, because it shows a significant decrease in the observable CEC values in all of the different curing times. Theoretically, alkalinity is one of the most significant factors that controls the crystallinity of zeolites. The increase in alkalinity is the reason for the increase in crystallinity, through nucleation and crystal growth as a result of the high concentration of reactive aluminates and silicates. There is a parallelism in the increase in alkalinity and the decrease in particle size, resulting in a slight distribution of particle size because of the increase in the rate of polymerization and nucleation. Therefore, similarly, smaller particle size could lead to a higher CEC.

In addition, the results show the importance of curing time in synthesizing zeolite A with a higher crystallinity and a smaller morphology, which may lead to higher CEC values. The curing time of 9 h showed a predominantly higher CEC value compared to the 6- and 12-h curing times. However, among all the CEC results of the 9-h curing time, the 1.0:1.5 ratio achieved the highest cation exchange capacity of 2.439 meq Na^+^/g of synthesized anhydrous zeolite A as compared to other fusion ratios. This result conforms with the data presented in Table 2. Theoretically speaking, zeolite A loses its cation exchange capacity over time, which is a sign that the zeolite’s negative charge was compensated partly by the adsorption of protons or any singly-charged or doubly-charged atoms from the solution, substituting one singly- or two singly-charged exchangeable atoms in the zeolite, respectively. The adsorption of these atoms into the zeolites is due to the greater selectivity of the zeolites toward them [90,91].

The commercial zeolite was also tested under the same conditions and resulted in a CEC performance of 1.641 meq Na^+^/g; this result shows that the CEC performance obtained from all the different fusion ratios of corn stover ash:NaOH under different curing times, as seen in Table 3, were more favorable than that of the commercial zeolite. Even the lowest CEC performance among all the fusion ratios and curing times of the corn stover ash:NaOH product, with a fusion ratio at 0.5:1.0 and a curing time of 6.0 h, showed a CEC performance of 1.789 meq Na^+^/g. The works of Qian and Li [43] support these results, in that the CEC value increases at first, then decreases with the rise in the NaOH concentration and curing time, singly. Furthermore, the rise in cation exchange capacity, as shown in Table 3, is congruent with the rise in crystallinity of the synthesized zeolite A from corn stover ash, as can be gleaned from Table 2. This result is of great importance in zeolite crystallinity since it has promising applications in removing sodium ions from seawater.

## 4. Conclusions

Zeolite A was successfully synthesized from corn stover ash as the raw material, then fused with NaOH as the basis of various fusion ratios.

Among the set of different temperatures that can be used to calcinate the corn stover ash, it was found, based on the results of the FT-IR, that 500 °C is an optimal temperature to calcinate the ash. This result supports the TGA-DTA that showed a range of 350 °C–550 °C is needed for the dehydroxylation of the structural OH of corn stover ash.

The fusion ratio of 1.0:1.5 of corn stover ash:NaOH showed the most favorable result among the different fusion rates. Based on SEM and XRD analysis, the morphology of the synthesized zeolite in this ratio was more favorable, compared to that of 1.0:2.0 corn stover ash:NaOH, despite the higher base concentration—this ratio resulted in the formation of a few traces of hydroxysodalite, which affected the CEC performance of the zeolite A.

The CEC performance also showed that the 1.0:1.5 ratio was the most favorable among the different fusion ratios, but among the different curing times, the 9-h curing time was the most favorable, yielding the highest CEC of 2.439 meq Na^+^/g of the synthesized anhydrous zeolite A. This curing time was also supported by the SEM results, showing more defined solids of cubic-rounded edge crystals. The CEC performance of all the different fusion ratios at different curing times of the corn stover ash:NaOH zeolite A was found to be more favorable than that of the commercial zeolite A (1.641 meq Na^+^/g).

These results highlight the process performed in order to obtain zeolite A from corn stover ash, which can be utilized for the removal of sodium ions from seawater. In addition, corn stover ash has great potential as an economic raw material for the industrial production of zeolite A. We propose that alleviating environmental problems brought forth by the rise in agricultural waste from corn crops can be easily accomplished by converting these wastes into zeolites. After all, zeolites are considered to be very useful materials in a vast variety of fields, e.g., as ion exchangers, among other applications. Interestingly, future research could evaluate the efficacy of zeolite A from corn stover ash in various cation exchange processes, as ion exchangers, adsorbents, and catalysts.

## Figures and Tables

**Figure 1 materials-14-04915-f001:**
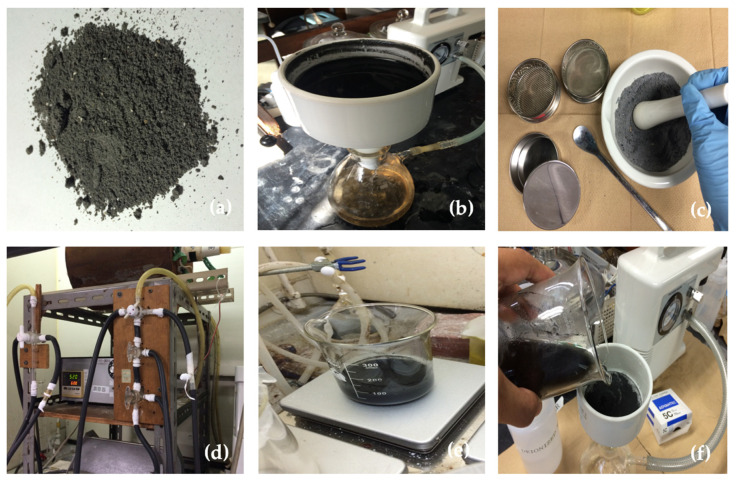
Corn stover ash from Ilocos Norte, Philippines: (**a**) as received; (**b**) the actual washing of corn stover; (**c**) the milling of corn stover; (**d**) the calcination of corn stover; (**e**) the acid treatment of corn stover; and, (**f**) the final washing of corn stover.

**Figure 2 materials-14-04915-f002:**
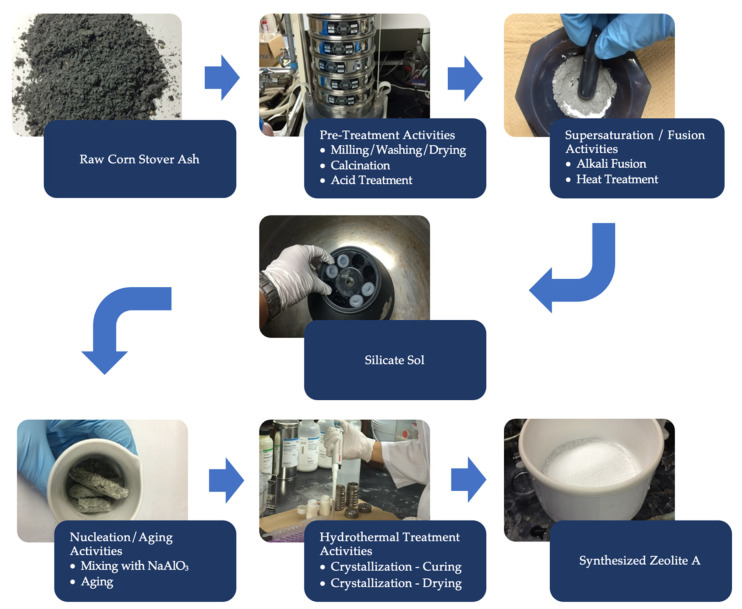
A schematic representation for the two-step zeolite A preparation method using corn stover ash.

**Figure 3 materials-14-04915-f003:**
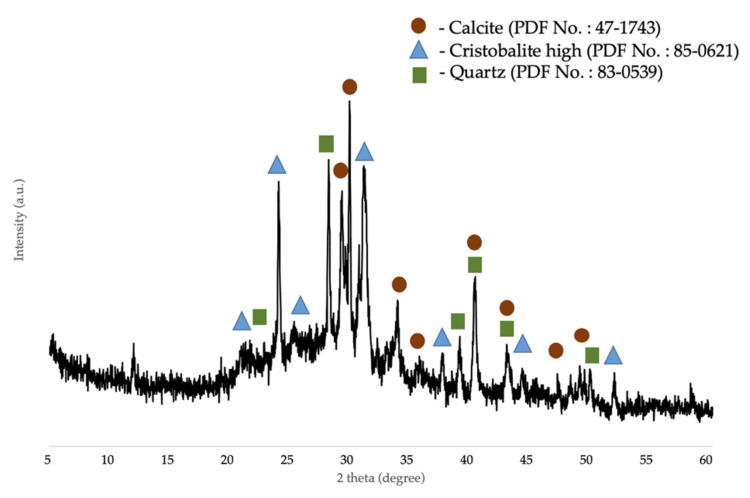
XRD patterns of raw corn stover ash (as received).

**Figure 4 materials-14-04915-f004:**
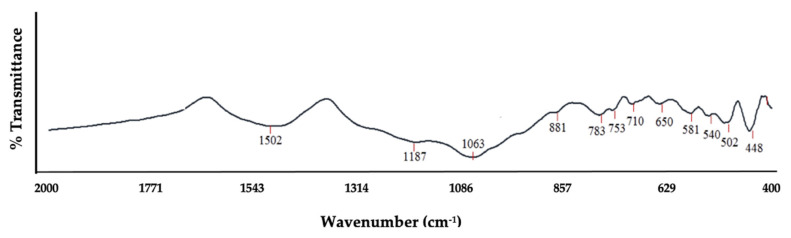
FT-IR spectra of raw corn stover ash (as received).

**Figure 5 materials-14-04915-f005:**
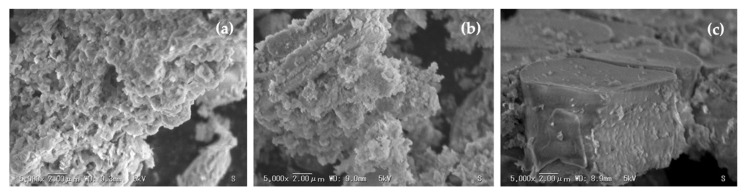
Morphologies of corn stover ash and related phases achieved by pretreatments. (**a**) Raw corn stover ash, as received, (**b**) calcined corn stover ash, and (**c**) acidified corn stover ash.

**Figure 6 materials-14-04915-f006:**
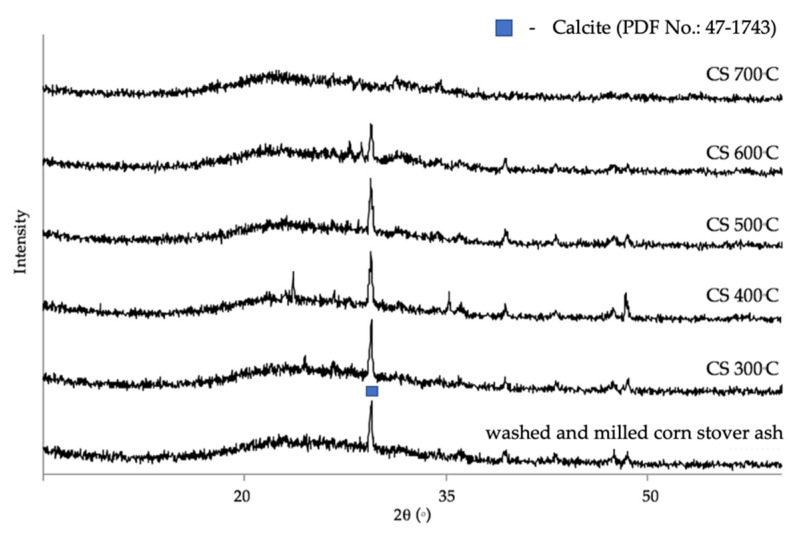
The XRD patterns of corn stover ash under different calcination temperatures.

**Figure 7 materials-14-04915-f007:**
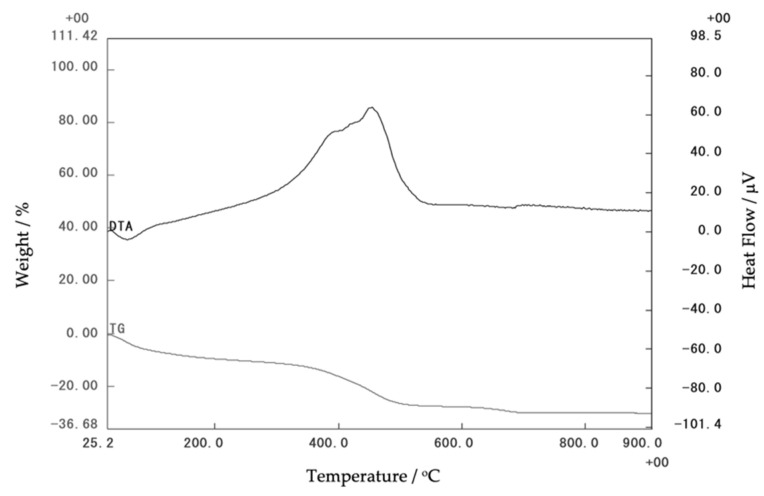
Thermogravimetric analysis of corn stover ash.

**Figure 8 materials-14-04915-f008:**
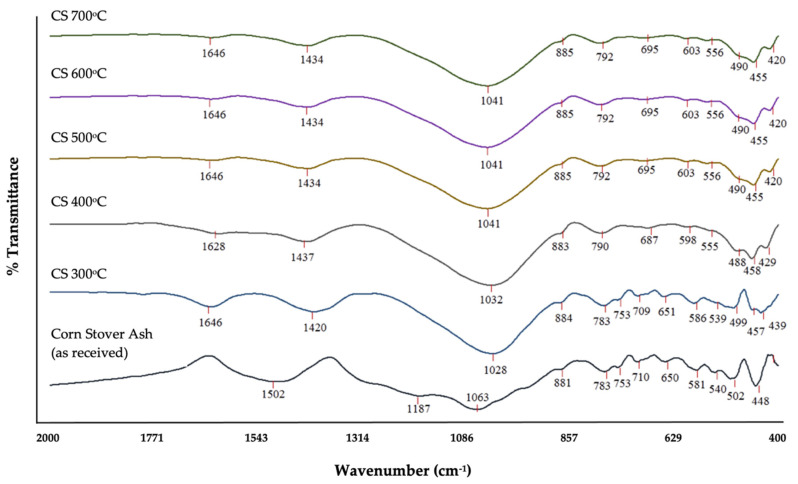
FT-IR spectra of corn stover ash under different calcination temperatures.

**Figure 9 materials-14-04915-f009:**
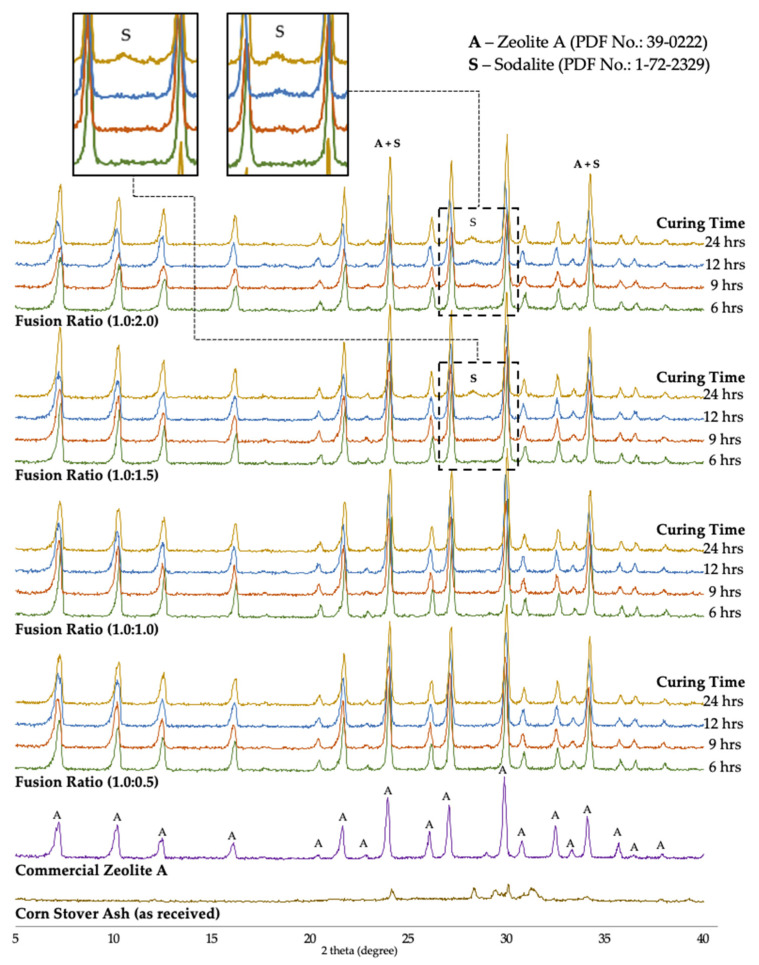
XRD patterns of raw corn stover ash (as received), commercial zeolite A, and synthesized zeolite A under different curing times achieved using different corn stover ash:NaOH ratios.

**Figure 10 materials-14-04915-f010:**
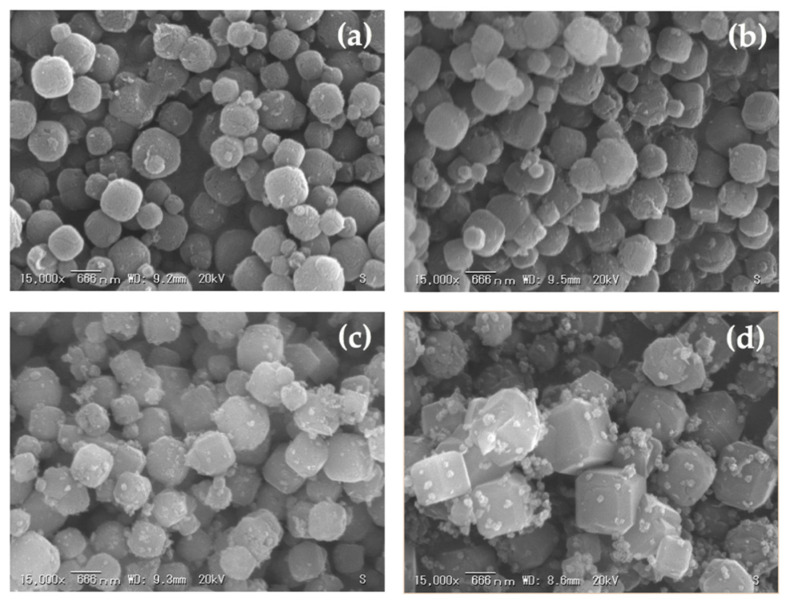
Morphologies of synthesized zeolite A with a curing time of 6.0 h and related phases achieved using different corn stover ash:NaOH ratios (**a**), corn stover ash:NaOH = 1:0.5, (**b**) corn stover ash:NaOH = 1:1.0, (**c**) corn stover ash:NaOH = 1:1.5, (**d**) corn stover ash:NaOH = 1:2.0.

**Figure 11 materials-14-04915-f011:**
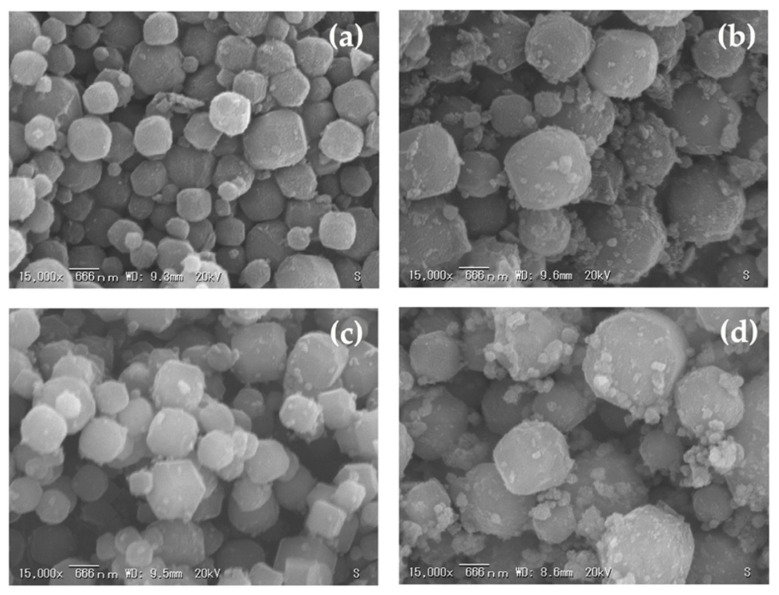
Morphologies of synthesized zeolite A with a curing time of 9.0 h and related phases achieved using different corn stover ash:NaOH ratios: (**a**) corn stover ash:NaOH = 1:0.5, (**b**) corn stover ash:NaOH = 1:1.0, (**c**) corn stover ash:NaOH = 1:1.5, (**d**) corn stover ash:NaOH = 1:2.0.

**Figure 12 materials-14-04915-f012:**
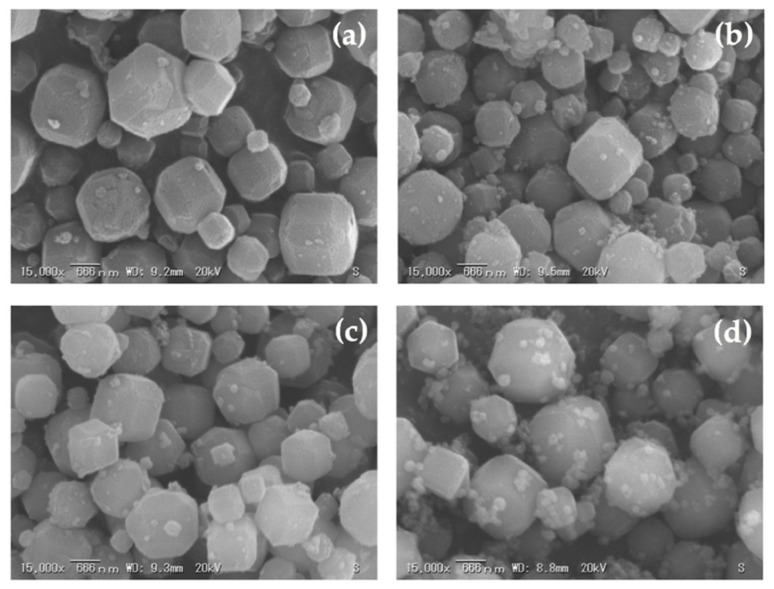
Morphologies of synthesized zeolite A with a curing time of 12.0 h and related phases achieved using different corn stover ash:NaOH ratios: (**a**) corn stover ash:NaOH = 1:0.5, (**b**) corn stover ash:NaOH = 1:1.0, (**c**) corn stover ash:NaOH = 1:1.5, (**d**) corn stover ash:NaOH = 1:2.0.

**Table 1 materials-14-04915-t001:** IR vibrational frequencies of corn stover ash.

Vibrations	Chemical Shift	Functionality	Absorption Bands (cm^−1^)
bending vibrations –O–T–O (deformation vibration of Si–O)			448, 502, 540
coordinated Al octahedral	C–H Bend (medium)	Alkene	581
symmetric stretching vibrations of bridge bonds –v_s_ T–O–T (v_s_ Si–O–Si)	Si–O	Silanol/Siloxane	650, 710, 753, 783
Si translation	C–H Bend (strong)	Aromatic	881
Bending vibration O–H		OH	1028
asymmetric stretching vibrations of bridge bonds –v_as_ T–O(T) (Si–O stretching)	C–O Stretch (strong)	Ether/Alcohol	1063, 1187

**Table 2 materials-14-04915-t002:** Percentage of crystallinity (%CXRD) of zeolite A synthesized using various curing times and fusion ratios.

Curing Time (h)	Fusion Ratios (Corn Stover Ash:NaOH) meq Na^+^/g			
	0.5:1.0	1.0:1.0	1.0:1.5	1.0:2.0
6.0	49.52%	52.38%	56.64%	51.26%
9.0	55.08%	55.97%	58.18%	52.30%
12.0	51.80%	52.73%	55.01%	50.50%
24.0	53.37%	55.16%	55.14%	53.07%

Note: %C_XRD_ of commercial zeolite A—53.51%.

**Table 3 materials-14-04915-t003:** Cation exchange capacities of zeolite A synthesized using various curing times and fusion ratios.

Curing Time (h)	Fusion Ratios (Corn Stover Ash:NaOH) meq Na^+^/g			
	0.5:1.0	1.0:1.0	1.0:1.5	1.0:2.0
6.0	1.789	1.852	1.925	1.849
9.0	1.827	2.142	2.439	2.275
12.0	1.861	2.139	2.428	2.158

Note: CEC of commercial zeolite A—1.641 meq Na^+^/g.

## Data Availability

The data underlying this article will be shared on reasonable request from the corresponding author.

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
