# Peer review of "Hydrothermal Synthesis and Characterization of Zeolite A from Corn (Zea Mays) Stover Ash"

_materials, 2021, doi:10.3390/ma14174915_

Round 1

Reviewer 1 Report

This paper by Pangan and coauthors described the synthesis of Zeolite A using waste by-products of maize agriculture. Pangan and coauthors have used several methods to characterize the raw product and the final product obtained during the treatment. The article is very interesting and presents results that may be of interest to materials scientists and the materials synthesis industry. However, as it stands it has severe shortcomings that need to be corrected before it can be accepted.

Therefore after major corrections, this works merits publication in Materials journal. Below, I have listed some general and specific comments that might be helpful for such revision.

General comments:

All the manuscript have to be improved, there are many typos and bad use of the English grammar.

Specific comments:

Page 5, section 3.1: All minerals composed by silicate tetrahedra are silicates (e.g. biotite, tourmaline, olivine, quartz, feldspar, etc…). Therefore, silicate is not an identification of the phase detected in the diffractogram. Which is the exact phase detected in the diffractogram? I searched the PDF 47-0715 in the ICSD database and there is not entry for such number. Could you please check that the number is also correct?

Page 5, lines 164 - 172: These sentences are repetitive and can be shortened. Due that in Fig. 3 the XRD pattern of the corn stover ash is identified with the peaks corresponding to the identified phases, it is not necessary to provide all values of 2theta in the main text.

Page 5, figure 3: There is two high peaks around the main calcite peak that are not been identified by the authors (there are also other minor peaks in the diffractogram which also deserves to be identified). Are these peaks corresponding to the two identified phases or correspond to new mineral phases? Moreover, from 70 to 90º there is not any interesting information. I suggest removing this part of the diffractogram to enlarge the image. By doing that, the authors will probably see that the unidentified peak around 33º are composed by two different peaks.

Page 6, lines 177-178: This sentence deserves some references to know exactly which are the references used by the authors to identified the IR spectra. In addition to table 1, I recommend to the authors to provide the IR spectra in a new figure, which can be used to be compared with the spectra shown in figure 7.

Page 7, line 205: This sentence would be valid if the authors had identified a phyllosilicate in the XRD, as they have not (and it does not appear to be a phyllosilicate because it lacks peaks in the 5º region) this sentence is not justified. The authors have to re-analyse the diffractogram and determine exactly which phase appears in order to make a structural analysis of this type.

Page 7, figure 5: Which phases appear and disappear during calcination? Why does the diffractogram of the initial product not coincide with the diffractogram of the initial product identified in figure 3? If they are not the same product, because they have received different treatments, the authors have to make this clear in the main text as well as in the figure caption (by giving different names to the initial products). If they are the same product, they have to explain in detail what the different peaks in the diffractograms depend on. On the X-axis I would recommend to make the scale every 5° to allow a better identification of the peaks.

Page 7, table 2: What is the reason for the different composition during calcination? It is not due to the phase change from crystalline to amorphous (a phase change does not produce a compositional change). The changes observed are so random that they are not due to decomposition of the product. The authors have to provide a plausible explanation for these variations or demonstrate the reasons for the variation in the composition of the product under study.

Pages 8-9, lines 238 – 290: These paragraphs are very repetitive without providing additional data. They can be summarised in a couple of paragraphs highlighting the main differences between the spectra.

Page 10, figure 8: Since the diffractogram of commercial zeolite A, which can be considered the standard, is given, it is not necessary to indicate which are the corresponding peaks of zeolite A in figure a. Why does the diffraction pattern of the initial product not match the one shown in figure 3? In figure d a peak corresponding to hydroxysodalite is identified. However, in the figure provided, the area where the peak is identified is indistinguishable from the background. The authors have to provide a better diffractogram, a zoom of that area or some additional test in which the presence of that phase in the diffractograms is confirmed beyond doubt. As long as they do not do so, the identification of this phase cannot be accepted.

Page 10, line 315: Zeolite A has been identified as CaCO3. Please, provide the exact composition of the zeolite A in the caption. Authors can also remove it because it is not necessary to provide the pattern as long as they are comparing their products with commercial zeolite A.

Page 11, lines 321-325: As stated above, authors have to prove that they obtained hydroxysodalite in their experiments. Figure 8 does not show that this phase appears in the product obtained.

Page 13, lines 388-390: Why do zeolites lose cation exchange capacity over time? This observation is worth discussing in more detail.

Page 14, lines 408 – 409: Please change the following sentence “This curing time is also supported by the XRD results…” to “… by the SEM results…”.

Page 14, lines 409 – 411: This sentence deserves to be commented in the results and discussion section. In section 3.5, there is not any comparison between the obtained zeolite and the commercial zeolite.

Author Response

Dear Sir/Madam:

This is to respectfully submit the changes I made based on your comments and observation. Please find the point-to-point response attached here.  Thank you for your kind observations that have made a huge impact on the improvement of our study. Rest assured that all the comments and observations were all taken into consideration May you find merit on these changes for this study’s acceptance for publication.

Thank you and more power to you!

Best regards,

Professor Norway J. Pangan

Reviewer 2 Report

The manuscript presented a novel method for the preparation of Zeolite A from Corn (Zea Mays) Stover Ash. This article cannot be accepted in its current form in Materials journal. Major revision have to be done.

1) A careful reading of the manuscript will lead the authors to correct some misprints and grammatical errors especially for the verb conjugation before submitting again.
2) In Figure 3 and Figure 5, why XRD patterns of raw corn stover ash are different?

3) In line 200, the authors claim that: “The calcined-acid treated corn stover ash honeycomb-like crystals disappeared and irregular nanoparticles formed.” But in Figure 4c, there are no nanoparticles.

4) In section 3.1, FT-IR spectra of corn stover ash in the literature is analyzed. Why not show the FT-IR spectra of corn stover ash in this work?

5) The authors stated on the basis of XRD patterns that “intensities of the sodalite peaks

in the pattern increased with the rise of NaOH concentration.” The authors should precise how they have calculated the crystallinity of the samples from XRD patterns, and give the calculated results.

6) In section 3.5, the authors use sodium acetate method to exchange all the cations in the material. However, Zeolite A originally contains abundant sodium ions for supplementing its skeleton charge. It is not feasible to exchange sodium ions in zeolite A with sodium acetate. The authors should adopt calcium ions or other cations to test the cation exchange capacity of Zeolite A.

Author Response

(The authors gave the same response as above.)

Reviewer 3 Report

This manuscript presents the synthesis and characterization of ion-exchangeable LTA zeolite from a natural silica source (i.e., corn stover ash). It is good to know that zeolite can be synthesized using abundant waste material obtained after corn cropping. However, there are so many reported routes to LTA and other zeolites from natural silica precursors, so there should be some interesting application properties or at least industrial advantage (e.g., cheap processing cost, easy silica extraction, and high purity of silica). I would not recommend for publication of this journal if not the raised issues (written below) are not clarified.

1. What is the distinct advantage of the method introduced in this manuscript? Is it cost-effective compared with using commercial sodium silicate (quite cheap), and other natural silica precursors? How good is the cation exchange capacity when compared with those of commercial LTA zeolites and laboratory-synthesized zeolites from commercial sodium silicate solution? These things should be discussed for claiming novelty.
2. There are emerging applications of LTA zeolites (https://doi.org/10.1016/j.micromeso.2019.109667), for example, adsorption (https://doi.org/10.3390/molecules25040944), and fluorescence (https://doi.org/10.1039/D1QM00299F). I think the information about the versatile application of LTA zeolites is more required for the introduction section rather than the information about the Philippines.
3. Can you explain why is the cation exchange capacity depending on both fusion ratios and curing time? Please specify one by one.
4. In most SEM images of LTA zeolites, there are relatively small particles (or impurities) on the relatively large LTA cubic-shaped crystals. What are the small things? Is there any possibility of elementally impure phases? There should be clear information about the elemental composition of extracted silica precursors used for zeolite synthesis.

Author Response

(The authors gave the same response as above.)

Round 2

Reviewer 1 Report

In this new version of the manuscript, the authors have improved all the points of weakness that were pointed out in my first review. Given that the manuscript has been significantly improved, I think it deserves to be published in Materials in its current state. 

Author Response

Dear Sir/Madam:

Thank you for your kind observations that have made a huge impact on the improvement of our study. Your approval for publication of this manuscript is highly appreciated.

Again, Thank you and More power!

Best regards,

Prof. Norway J Pangan

Reviewer 2 Report

The manuscript presented a novel method for the preparation of Zeolite A from Corn (Zea Mays) Stover Ash. This article cannot be accepted in its current form in Materials journal. Major revision have to be done.

1) A careful reading of the manuscript will lead the authors to correct some misprints and grammatical errors especially for the verb conjugation before submitting again.
2) In Figure 3 and Figure 5, why XRD patterns of raw corn stover ash are different?

3) In line 200, the authors claim that: “The calcined-acid treated corn stover ash honeycomb-like crystals disappeared and irregular nanoparticles formed.” But in Figure 4c, there are no nanoparticles.

4) In section 3.1, FT-IR spectra of corn stover ash in the literature are analyzed. Why not show the FT-IR spectra of corn stover ash in this work?

5) The authors stated on the basis of XRD patterns that “intensities of the sodalite peaks

in the pattern increased with the rise of NaOH concentration.” The authors should precise how they have calculated the crystallinity of the samples from XRD patterns, and give the calculated results.

6) In section 3.5, the authors use sodium acetate method to exchange all the cations in the material. However, Zeolite A originally contains abundant sodium ions for supplementing its skeleton charge. It is not feasible to exchange sodium ions in zeolite A with sodium acetate. The authors should adopt calcium ions or other cations to test the cation exchange capacity of Zeolite A.

Author Response

Dear Sir/Madam:

This is to respectfully submit the changes I made based on your comments and observations. Please find the point-to-point response attached here. May you find merit on these changes for this study's acceptance for publication.

Thank you and more power to you!

Best Regards,

Prof. Norway Pangan

Reviewer 3 Report

The authors have answered my comments in this revision round. However, these works trying a variation of synthesis precursors have already overwhelmingly been published in the field of materials science. Apparently, there is no new dramatic advance in zeolite science. There should have been some novel application results in the revised manuscript, rather than introducing other researchers' recent advances. In short, how is the practical impact of the natural precursor-derived zeolite A compared with commercial zeolite A in actual ion-exchange applications (e.g., radioactive Cs removal, specific ion removal in seawater as the author stated)?

Author Response

Dear Sir/Madam:

This is to respectfully submit the changes I made based on your comments and observations. Please find the response attached here. Thank you for your kind observations that have made a huge impact on the improvement of our study.

May you find merit on these changes for this study's acceptance for publication

Thank you and More power to you!

Best Regards,

Prof. Norway J Pangan

Round 3

Reviewer 2 Report

The paper has certain novelty and advantages for this field research work,so I suggest this manuscript can be published in Materials.

Reviewer 3 Report

The reviewer had asked the authors to demonstrate ion-exchange applicability for practical purposes. The authors' report shows that there is no practical example in the revised manuscript. They are insisting that they are doing other research, but the work submitted to this journal still does not lack in showing fascinating application properties, considering so many papers about zeolite synthesis with natural silica (or alumina) sources. Moreover, the reviewer does not think that the use of another precursor fits the aims and scope of this journal Materials.